# Clinical Presentations and Multimodal Imaging Diagnosis in Chronic Thromboembolic Pulmonary Hypertension

**DOI:** 10.3390/jcm11226678

**Published:** 2022-11-10

**Authors:** Mi-Hyang Jung, Hae Ok Jung, Soo Jin Kwon, Suyon Chang

**Affiliations:** 1Division of Cardiology, Department of Internal Medicine, Seoul St. Mary’s Hospital, College of Medicine, The Catholic University of Korea, Seoul 06591, Korea; 2Catholic Research Institute for Intractable Cardiovascular Disease, College of Medicine, The Catholic University of Korea, Seoul 06591, Korea; 3Division of Nuclear Medicine, Department of Radiology, Seoul St. Mary’s Hospital, College of Medicine, The Catholic University of Korea, Seoul 06591, Korea; 4Department of Radiology, Seoul St. Mary’s Hospital, College of Medicine, The Catholic University of Korea, Seoul 06591, Korea

**Keywords:** CTEPH, multimodal imaging, diagnosis, hypertension, pulmonary

## Abstract

Chronic thromboembolic pulmonary hypertension (CTEPH) is a rare but life-threatening pulmonary vascular disease caused by the presence of a prolonged thrombus in the pulmonary artery. CTEPH is a distinct disease entity classified as group 4 pulmonary hypertension according to the World Symposium on Pulmonary Hypertension. It is the only potentially curable cause of pulmonary hypertension. However, timely diagnosis and treatment are often hampered by nonspecific symptoms and signs and a lack of physician awareness regarding the condition. Thus, it is important to be familiar with the clinical features of CTEPH and the associated diagnostic processes. Herein, we cover the diagnostic approach for CTEPH using multimodal imaging tools in a clinical setting.

## 1. Introduction

Chronic thromboembolic pulmonary hypertension (CTEPH) is a rare but life-threatening pulmonary vascular disease caused by the presence of a prolonged thrombus in the pulmonary artery (PA). Chronically organized in situ thrombi that have not been dissolved after previous venous thromboembolic events trigger various pulmonary vascular pathologies, such as hypertrophy, remodeling, stenosis, and occlusion, which lead to pulmonary arterial hypertension and eventually right-heart failure (Figure 1). Currently, CTEPH is classified as a group 4 pulmonary hypertension (PH) based on the 6th World Symposium on Pulmonary Hypertension and recent PH guidelines [1,2]. Classically, CTEPH can be confirmed when the mean pulmonary arterial pressure (mPAP) is greater than 25 mmHg and the pulmonary artery wedge pressure is less than or equal to 15 mm Hg measured via right-heart catheterization (RHC), combined with a mismatched perfusion defect with obvious evidence of multiple chronic or organized thrombi or emboli in the pulmonary arteries of the lung, despite at least 3 months of effective anticoagulation [1]. In this setting, PH might be caused by PA obstruction by an organized thrombus and associated microvasculopathy [3]. Recently, the European Society of Cardiology and the European Respiratory Society lowered the diagnostic criteria for precapillary PH (mPAP from 25 to 20 mmHg and pulmonary vascular resistance from 3 to 2 Wood units), enabling the diagnosis of CTEPH earlier in the course of the disease [2]. 

The risk factors for CTEPH appear to differ from those for acute pulmonary embolism (PE), such as immobilization or recent surgery. The potential risk factors for CTEPH include certain chronic medical conditions (e.g., permanent intravascular devices, inflammatory bowel diseases, autoimmune disease, hypothyroidism, splenectomy, and malignancy), thrombophilia, and genetic predisposition. However, many cases of CTEPH develop without previous acute PE [4,5,6,7]. The rarity of the disease, nonspecific symptoms and signs, and a lack of physicians’ awareness of CTEPH (including when to suspect it and how to evaluate it) are barriers to timely diagnosis [8]. Therefore, despite the presence of effective therapy (pulmonary endarterectomy [PEA]), many patients are diagnosed only at the late stage of the disease, when distal PA obstruction and microvasculopathy have already progressed. Patients with advanced disease are not eligible for PEA because the procedure can only treat proximal lesions. Thus, a high index of suspicion and timely diagnosis using multimodal imaging tools is mandatory for optimal patient outcomes. In this review, we describe how to approach suspected CTEPH through a clinical vignette and discuss the role of multimodal imaging in diagnosing CTEPH.

## 2. Clinical Presentation: Learning from a Case

A 46-year-old female visited our hospital with worsening exertional dyspnea. Three years ago, she was diagnosed with an acute PE and prescribed warfarin for 18 months. However, 1 year after stopping the warfarin, the patient’s dyspnea worsened. Edema in both lower legs developed 1 month before hospitalization. She visited the local clinic, where she was diagnosed with recurrent PE, and was subsequently transferred to our hospital. Despite receiving an adequate dose of warfarin for 3 months, she still complained of dyspnea. Follow-up echocardiography revealed a D-shaped left ventricle (LV) with an elevated right ventricular systolic pressure of 62 mmHg. No evidence of significant left-sided heart disease was found. Computed tomography (CT) imaging showed a dilated main PA but no evidence of PE or lung parenchymal disease (Figure 2). What should be done for further work-up of this patient?

CTEPH should be suspected, especially in patients with acute PE who have persistent PH or do not recover from acute PE. However, the absence of prior acute PE events does not preclude the possibility of CTEPH [9,10]. In the early stages of the disease, patients might not have any symptoms. Even if patients experience symptoms, they can be nonspecific symptoms such as dyspnea and fatigue that can occur in patients with PAH, acute PE, or other pulmonary diseases. Similar to other forms of PH, the most common symptoms of CTEPH are exercise intolerance and exertional dyspnea, which are attributable to limited cardiac output and increased dead space ventilation [4]. As the disease worsens and right ventricular dysfunction progresses, other symptoms such as lower leg swelling, abdominal discomfort, chest pain, hemoptysis, and dizziness with or without syncope could occur. Currently, routine surveillance for CTEPH is not recommended in asymptomatic patients with a history of acute PE, given the low incidence of CTEPH after acute PE [11]. As shown in the index case, some people might present predominantly with microvasculopathy (changes in the distal pulmonary vascular bed) with near resolution of the central pulmonary thrombus, which often complicates the establishment of an accurate diagnosis. In this case, the CT scan demonstrated small segmental PA diameters and sparse distal branches, which are typical findings of CTEPH microvasculopathy (Figure 2). However, in practice, this can be overlooked without experience or a high index of suspicion. Histological changes in microvasculopathy include eccentric intimal fibrosis, thickening, and diffuse distal thrombosis [12,13,14].

## 3. Diagnosis (Using Multimodal Imaging)

In this section, we discuss the utility of multimodal imaging for the diagnosis of CTEPH. Echocardiography, CT, and ventilation-perfusion (V/Q) scans can be utilized for suspected PH. The presence of a V/Q mismatch in the setting of PH should prompt further evaluation using RHC and pulmonary angiography. Each imaging modality has its own role; thus, comprehensive evaluation using multimodal imaging is crucial for the proper diagnosis and management of patients with CTEPH (Table 1).

### 3.1. Echocardiography

Echocardiography is the preferred imaging modality for the initial screening and follow-up of CTEPH. It facilitates the estimation of systolic PA pressure, comprehensive evaluation of right ventricle (RV)/ right atrium (RA) size and function, and identification of other causes of PH (such as PH due to congenital heart disease or left heart disease) [2,16,17]. However, one should be cognizant that echocardiography alone is insufficient to confirm PH, which should be confirmed through RHC.

The typical echocardiographic findings for PH (regardless of etiology) are shown in Figure 3. The degree of RV enlargement can be visually assessed using the parasternal long-axis view and apical four-chamber view (Figure 3A). The flattened interventricular septum, called the D-shaped LV, suggests RV pressure overloading (Figure 3B). In some PH cases, a midsystolic notch and accelerated RV outflow tract time can represent elevated PA pressure (Figure 3C). Through data on tricuspid regurgitation (TR) Vmax and inferior vena cava (IVC) status, systolic PA pressure can be estimated in the absence of pulmonary stenosis (Figure 3D). Recent European guidelines for PH recommend the use of TR Vmax rather than the estimated value of PA pressure, given the possible errors (e.g., inaccurate RA pressure estimates based on IVC status and errors stemming from the amplification of TR Vmax) [2]. A TR Vmax of 2.8 m/s or greater may suggest PH. However, TR Vmax also has many limitations; thus, the presence and degree of PH should be interpreted in the context of other echocardiographic findings. Pericardial effusion can be observed in advanced cases of CTEPH (Figure 3A,B, marked with asterisks). Although the precise mechanism for pericardial effusion is not clear, markedly elevated RV and RA pressures are thought to be one possible reason that detrimentally affects lymphatic and venous drainage from the pericardium [18]. However, echocardiography has a limited role in the assessment of PH etiologies, particularly in relation to lung parenchymal disease or pulmonary vessels.

RV function assessment is another essential part of the initial and serial evaluation of CTEPH [19,20,21,22,23,24,25]. Decreased tricuspid annular plane systolic excursion (TAPSE, <18 mm), RV fractional area change (FAC, <35%), and tricuspid annulus velocity (s’ < 9.5 cm/s) are indicative of RV dysfunction [19,20]. Considering the complex geometry of the RV, recent echocardiographic techniques (such as 3D volume imaging and/or RV strain) could provide more accurate information.

### 3.2. CT Scan

CT scans provide valuable information regarding the presence and possible etiologies of PH [15,26,27,28,29]. During the CTEPH evaluation process, CT findings suggestive of PH and the presence of a chronic organized thrombus should be meticulously examined.

#### 3.2.1. CT Findings Suggestive of PH

Representative CT findings for PH included an increased PA diameter (PA to aorta ratio > 1, PA diameter > 29 mm), straightening or leftward bowing of the interventricular septum, RV dilatation (RV:LV diameter ratio ≥ 1 at the midventricular level on axial images), and hypertrophy (Figure 4).

#### 3.2.2. CT Findings for Chronic Thrombus: Morphology and Location

Contrast-enhanced CT also provides information regarding the thrombus morphology and location. In particular, CT helps in the differentiation of thrombus in the acute versus chronic stages (Figure 5 and Figure 6). Eccentric filling defects (thrombus adhering to the vessel wall), webs or bands in the PA, or calcified thrombus are features of chronic clots and reflect the organization and partial recanalization of a previous acute thrombus [2,29,30]. Acute clots usually present as central filling defects and occur at branch points (Figure 5B) [29]. Moreover, pulmonary vessel characteristics (retraction, atrophy, and poststenotic dilatation) could also provide clues to their chronicity. Sometimes, chronic complete vascular occlusion may be presented as a convex-shaped distal vessel cutoff, the so-called “pouch defect” on pulmonary angiography [31]. Mosaic attenuation and enlarged bronchial arteries are other indicators of CTEPH. Mosaic attenuation is a term that describes heterogeneous attenuation of the lung parenchyma on CT. In mosaic attenuation related to the PH, the area of decreased attenuation indicates a hypoperfused lung due to vascular obstruction; conversely, the normal attenuated region suggests normal perfusion (Figure 6C). However, mosaic attenuation is a nonspecific finding that can be found in other diseases, including small airway disease and small vessel disease. The enlarged bronchial artery (collateral vessel formation) also strongly suggests the diagnostic possibility of CTEPH [32], considering the dual blood supply to the lung via the pulmonary and bronchial arteries.

CT also provides information on the location of the thrombus, which is essential for selecting therapeutic options. Surgical PEA is the treatment of choice for CTEPH because it is potentially curative. However, when surgery seems to be impossible or ineffective, usually in cases of distal segmental and subsegmental PA with webs and slit-like lesions, balloon pulmonary angiography might be considered [2]. However, the diagnostic accuracy of CT for CTEPH is relatively low, especially at the subsegmental level, with a sensitivity and specificity of 76% and 96%, respectively [33]. Moreover, its diagnostic accuracy may be much lower in real-world settings, especially in low-volume institutions, because of the rarity of the disease [34]. However, recent innovative techniques can improve the sensitivity of the diagnosis of subsegmental PA lesions. Dual energy CT (DECT) can create a perfused blood volume (PBV) map. PBV maps help detect perfusion defects, even in the absence of vascular stenotic findings, especially at the subsegmental level (Figure 7) [29,35].

### 3.3. V/Q Scan

Currently, the V/Q scan remains the most effective imaging modality for excluding CTEPH [2,36,37,38]. The interpretation of the V/Q scan is typically straightforward. Moreover, it has high sensitivity (90–100%) and specificity (94–100%) [37]. Most patients with CTEPH have abnormal V/Q scan images with multiple moderate and large perfusion defects that can be easily identified (Figure 8) [38,39,40]. Although some recent studies using novel CT techniques have shown comparable sensitivity and specificity of V/Q scans and CT [29,41], they are expensive, more technically challenging, and have limited availability for wide adaptation [2]. Single-photon emission computed tomography (SPECT) V/Q scan is gaining increasing attention owing to its enhanced sensitivity and specificity compared to planar imaging [42].

### 3.4. RHC and Digital Subtraction Angiography (DSA)

In suspected cases of CTEPH, RHC and DSA should be performed for confirmatory diagnosis (Figure 9). RHC can confirm the presence and degree of precapillary PH, and DSA can accurately identify the affected PA vessels. These are mandatory steps for the final diagnosis of CTEPH.

Classically, CTEPH can be confirmed when the mPAP is greater than 25 mmHg and the pulmonary artery wedge pressure is less than or equal to 15 mmHg measured via RHC [1]. However, as mentioned in the Introduction, recent European guidelines for PH have lowered the diagnostic threshold for PH (mPAP from 25 to 20 mmHg and pulmonary vascular resistance from 3 to 2 Wood units) [2]. We hope that this revision will lead to diagnosis earlier in the course of the disease and result in the effective management of CTEPH.

## 4. Conclusions

In summary, CTEPH is a rare, fatal, and disabling disease that requires timely diagnosis and treatment. For a timely diagnosis, a high index of suspicion and additional work-up are warranted for patients with unexplained dyspnea or signs compatible with right-sided heart failure. Comprehensive evaluation using multimodal imaging tools (echocardiography, CT, and V/Q scan) is crucial for enhancing the diagnostic yield of CTEPH. In suspected cases of CTEPH, RHC and DSA are mandatory for the final diagnosis and decision regarding therapeutic options.

## Figures and Tables

**Figure 1 jcm-11-06678-f001:**
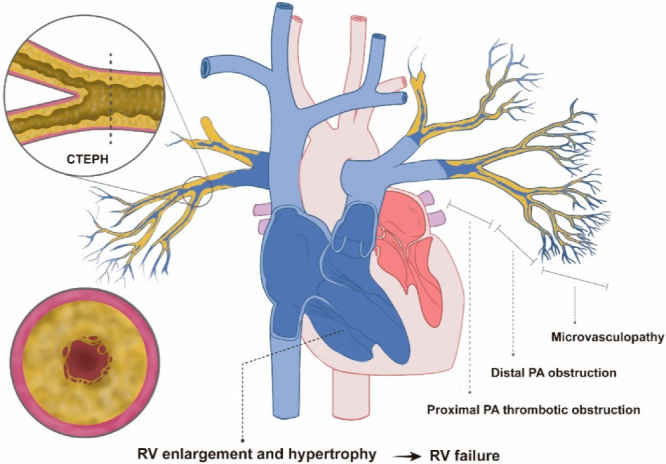
Pathophysiology of CTEPH. Abbreviations: CTEPH, chronic thromboembolic pulmonary hypertension; PA, pulmonary artery; RV, right ventricle.

**Figure 2 jcm-11-06678-f002:**
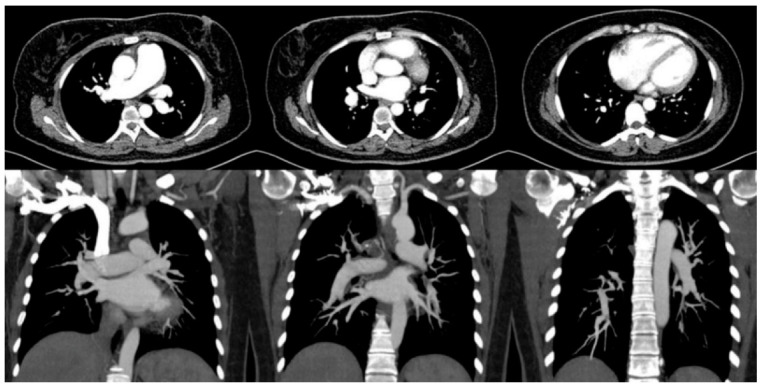
CT findings in the case. CT imaging showed a dilated main PA and a dilated RV but no evidence of PE or lung parenchymal disease.

**Figure 3 jcm-11-06678-f003:**
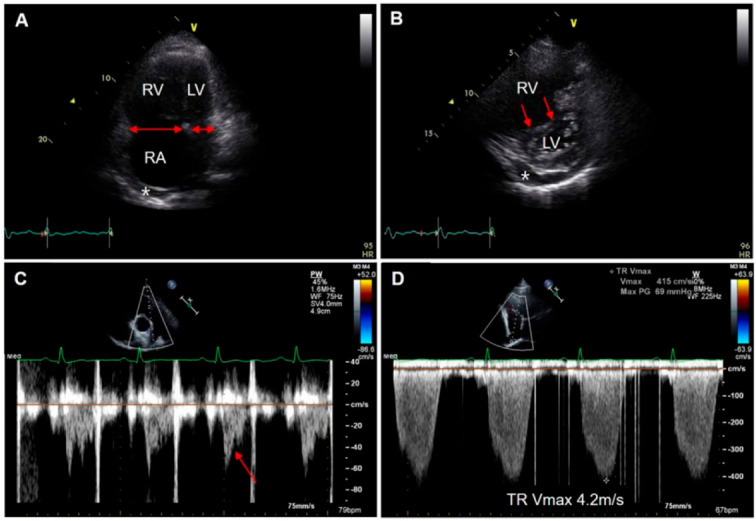
Echocardiographic findings associated with CTEPH. Note the dilated RV (**A**) and flattened interventricular septum (called the D-shaped LV, red arrows), which suggest RV pressure overloading (**B**). Pericardial effusion (asterisk) can be observed in advanced cases of CTEPH (**B**). Pulsed-wave Doppler image obtained from the RV outflow tract shows a midsystolic notch (arrow) and shortened acceleration time, which reflect increased pulmonary vascular resistance (**C**). Continuous-wave (CW) Doppler analysis at the tricuspid valve reveals a TR Vmax of 4.2 m/s, which strongly suggests significant PH (**D**).

**Figure 4 jcm-11-06678-f004:**
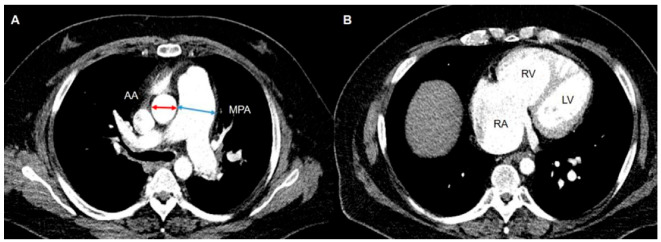
CT findings suggestive of PH. (**A**) CT demonstrates a dilatation of the main pulmonary artery (MPA). The ratio of the MPA diameter to that of the ascending aorta (AA) is greater than 1. (**B**) CT scan obtained at a lower level shows RV dilatation. The ratio of the RV diameter to the LV diameter is greater than 1 at the midventricular level. Abbreviations: AA, ascending aorta; LV, left ventricle; MPA, main pulmonary artery; RA, right atrium; RV, right ventricle.

**Figure 5 jcm-11-06678-f005:**
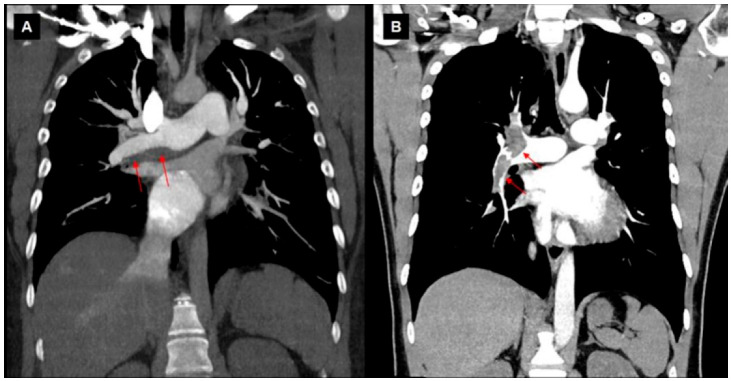
Distinct thrombus morphology for chronic (**A**) and acute thrombus (**B**). Note the eccentric thrombus (adhering to the vessel wall) in CTEPH (**A**); central filling defect in acute PE.

**Figure 6 jcm-11-06678-f006:**
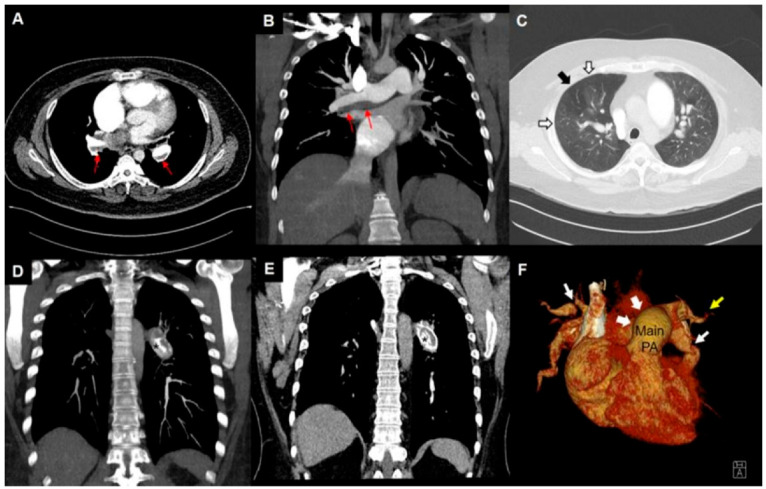
CT findings suggestive of CTEPH. Eccentric thrombus along the wall (**A**,**B**). Mosaic attenuation (**C**). Calcified thrombus (asterisks, (**D**,**E**)). Three-dimensional volume rendering image showing dilated main PA (arrowhead) and narrowing (white arrow) and pruning (yellow arrow) of PA (**F**).

**Figure 7 jcm-11-06678-f007:**
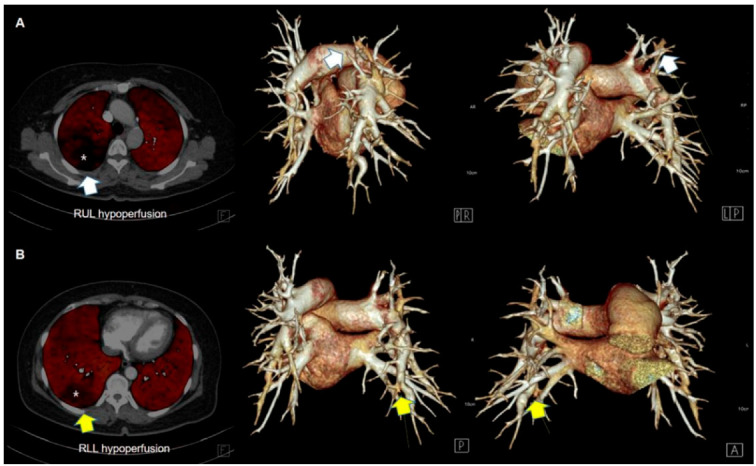
PBV maps and corresponding 3D volume rendering images of pulmonary arteries in a CTEPH patient. PBV maps demonstrate perfusion defects (regions marked with an asterisk) in the right upper lobe (RUL) (**A**) and right lower lobe (RLL) (**B**). Three-dimensional volume rendering images demonstrate attenuation and occlusion of corresponding vessels.

**Figure 8 jcm-11-06678-f008:**
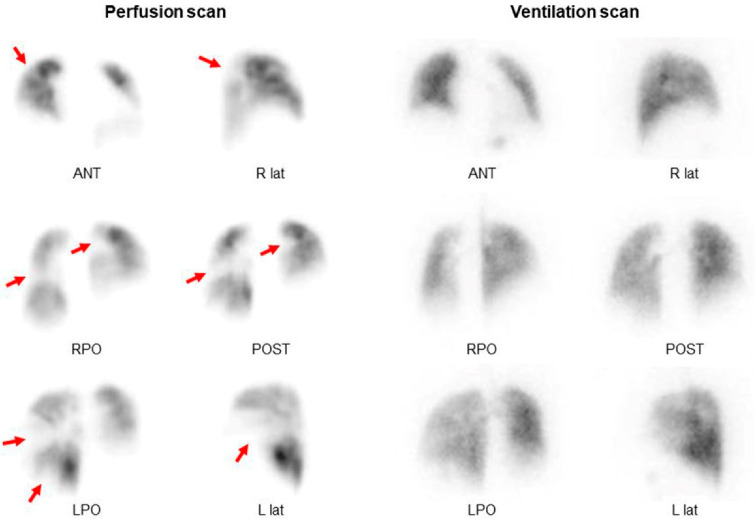
V/Q scan findings demonstrating multiple perfusion defects in both lungs. Arrows indicate areas of showing perfusion defects. Abbreviations: ANT, anterior; R lat, right lateral; RPO, right posterior-oblique; POST, posterior; LPO, left posterior-oblique; L lat, left lateral.

**Figure 9 jcm-11-06678-f009:**
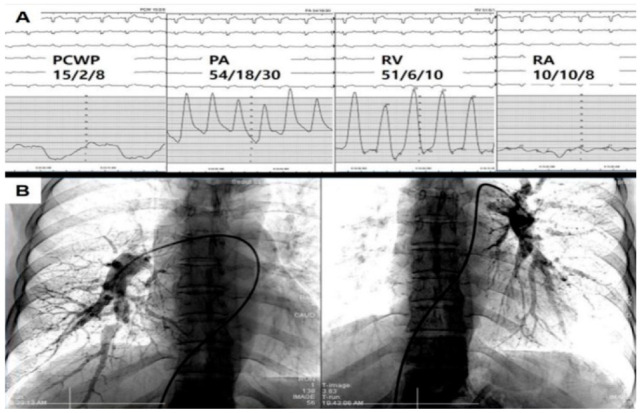
The final diagnosis can be made via RHC (**A**) and DSA (**B**). Abbreviations: PCWP, pulmonary capillary wedge pressure.

**Table 1 jcm-11-06678-t001:** Strengths and limitations of imaging modalities.

Variable	Echocardiography	CT Scan	V/Q Scan
PH detection	++	+	−
Assessment of PH etiology	++	+++	++
Evaluated anatomic region			
Lung parenchyma	−	+++	−
Cardiac chambers	++	++ (especially on ECG-gated CT)	−
Pulmonary vessels	+	+++	+
Strengths	Good for initial screening and follow-up for PH; readily available; safe for repeated testing	Excellent evaluation for etiologies of PH	Good for initial screening of thromboembolic disease (acute PE, CTEPH)
Weaknesses	Limited role for the assessment of etiology (lung parenchyma, pulmonary vessels); cannot confirm PH and requires RHC	Radiation risks; limited assessment of hemodynamic assessment; needs experienced radiologists; cannot confirm PH and requires RHC	Findings often nonspecific; needs additional testing to assess etiology; cannot confirm PH and requires RHC

Abbreviations: CT, computed tomography; CTEPH, chronic thromboembolic pulmonary hypertension; ECG, electrocardiogram; PE, pulmonary embolism; PH, pulmonary hypertension; RHC, right-heart catheterization; V/Q, ventilation/perfusion. Modified from reference [15]. − = unavailable; + = limited use; ++ = moderately useful; +++ = very useful.

## Data Availability

Not applicable.

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
