# Peer review of "Clinical Presentations and Multimodal Imaging Diagnosis in Chronic Thromboembolic Pulmonary Hypertension"

_jcm, 2022, doi:10.3390/jcm11226678_

Round 1

Reviewer 1 Report

Thank you for submitting this original paper to Journal of Clinical Medicine. It’s a pleasure for me to receive this manuscript as a reviewer.

In this review, you summarize the clinical presentation and the diagnosed of the Chronic Thromboembolic Pulmonary Hypertension (CTEPH) that is still a rare and misdiagnosed pathology.

I have some comments and suggestions:

  1. The introduction is well presented: in a concise and precise modality, all the risk factors and presentation symptoms are described.
  2. I think the clinical presentation starting from a concrete clinical case it is very interesting and innovative.
  3. The diagnosed section is complete and well detailed.
  4. I think that at line 207 the reference to the PEA and in general to the surgical treatment is too short. In the last guidelines (ESC 2022 PH), it is absolutely specified that PEA is the treatment of choice for CTEPH and, in referred centre, it could treat segmental and subsegmental obstruction (different from that you have reported at line 208). I suggest bettering explaining this part to give the more complete information in line with the rest of the manuscript.
  5. The figures adopted are very interesting
  6. In general, the manuscript is well writing.

Thank again for submitting this paper.

Author Response

Reviewer #1:

Thank you for submitting this original paper to Journal of Clinical Medicine. It’s a pleasure for me to receive this manuscript as a reviewer.

In this review, you summarize the clinical presentation and the diagnosed of the Chronic Thromboembolic Pulmonary Hypertension (CTEPH) that is still a rare and misdiagnosed pathology.

I have some comments and suggestions:

  1. The introduction is well presented: in a concise and precise modality, all the risk factors and presentation symptoms are described.

Response: We would like to thank the reviewer for evaluating our manuscript and for the positive comment.

  1. I think the clinical presentation starting from a concrete clinical case it is very interesting and innovative.

Response: We would like to thank the reviewer for the positive comment.

  1. The diagnosed section is complete and well detailed.

Response: We would like to thank the reviewer for the positive comment.

  1. I think that at line 207 the reference to the PEA and in general to the surgical treatment is too short. In the last guidelines (ESC 2022 PH), it is absolutely specified that PEA is the treatment of choice for CTEPH and, in referred centre, it coud treat segmental and subsegmental obstruction (different from that you have reported at line 208). I suggest bettering explaining this part to give the more complete information in line with the rest of the manuscript.

Response: We would like to thank the reviewer for the insightful comments. We agree with the reviewer’s comment that PEA is the only curative and preferred therapy for CTEPH. According to the reviewer’s comment, we have revised the relevant statement in the manuscript. Unfortunately, our topic was confined to clinical presentations and multimodal imaging diagnosis. Therefore, the treatment was not described in detail in the current manuscript. To our knowledge, other authors in this Special Issue regarding Chronic Thromboembolic Pulmonary Hypertension have already described it in detail. We hope that our revision will meet the reviewer’s expectations. The added part is as follows:

“Surgical PEA is the treatment of choice for CTEPH because it is potentially curative. However, when surgery seems to be impossible or ineffective, usually in cases of distal segmental and sub-segmental PA with webs and slit-like lesions, balloon pulmonary angiography might be considered [2].”

  1. The figures adopted are very interesting.

Response: We would like to thank the reviewer for the positive comment.

  1. In general, the manuscript is well writing.

Response: We would like to thank the reviewer for the positive comment.

Reviewer 2 Report

This review is well written and compact for medical staff who is dealing with CTEPH and pulmonary endarterectomy. It is also easy to read. Congratulations to authors!

I really congratulate the authors for their review paper on Chronic thromboembolic pulmonary hypertension (CTEPH). They systematically reviewed the diagnostic tools and treatment options in this difficult disease presenting this with a case. Especially they presented all the critical issues using diagnostic technologies in this area. They also discussed the strengths and limitations of imaging modalities. This didactically well-designed review will help the medical staff working on CTEPH a very important literature.

Author Response

This review is well written and compact for medical staff who is dealing with CTEPH and pulmonary endarterectomy. It is also easy to read. Congratulations to authors!

I really congratulate the authors for their review paper on Chronic thromboembolic pulmonary hypertension (CTEPH). They systematically reviewed the diagnostic tools and treatment options in this difficult disease presenting this with a case. Especially they presented all the critical issues using diagnostic technologies in this area. They also discussed the strengths and limitations of imaging modalities. This didactically well-designed review will help the medical staff working on CTEPH a very important literature.

Response: We would like to thank the reviewer for evaluating our manuscript and for the positive comment.